# Current Drug Development Overview: Targeting Voltage-Gated Calcium Channels for the Treatment of Pain

**DOI:** 10.3390/ijms24119223

**Published:** 2023-05-25

**Authors:** Flavia Tasmin Techera Antunes, Maria Martha Campos, Vanice de Paula Ricardo Carvalho, Claudio Antonio da Silva Junior, Luiz Alexandre Viana Magno, Alessandra Hubner de Souza, Marcus Vinicius Gomez

**Affiliations:** 1Department of Physiology and Pharmacology, University of Calgary, Calgary, AB T2N 1N4, Canada; flaviatasmim.techera@ucalgary.ca; 2Hotchkiss Brain Institute, University of Calgary, Calgary, AB T2N 1N4, Canada; 3Programa de Pós-Graduação em Odontologia, Escola de Ciências da Saúde e da Vida, Pontifícia Universidade Católica do Rio Grande do Sul, Porto Alegre 90619-900, RS, Brazil; maria.campos@pucrs.br; 4Faculdade Santa Casa BH, Belo Horizonte 30110-005, MG, Brazil; vanice.carvalho@feluma.org.br (V.d.P.R.C.); claudiojunior.biologia@gmail.com (C.A.d.S.J.); 5Programa de Pós-Graduação em Ciências da Saúde, Faculdade Ciências Médicas de Minas Gerais (FCMMG), Belo Horizonte 30110-005, MG, Brazil; luiz.magno@cienciasmedicas.edu.br (L.A.V.M.); alessandra.souza@cienciasmedicasmg.edu.br (A.H.d.S.)

**Keywords:** analgesic, antinociception, calcium signaling, drug development, pain transmission

## Abstract

Voltage-gated calcium channels (VGCCs) are targeted to treat pain conditions. Since the discovery of their relation to pain processing control, they are investigated to find new strategies for better pain control. This review provides an overview of naturally based and synthetic VGCC blockers, highlighting new evidence on the development of drugs focusing on the VGCC subtypes as well as mixed targets with pre-clinical and clinical analgesic effects.

## 1. Introduction

The high worldwide prevalence of pain and diseases associated with pain cause disability and reduced quality of life, with negative impacts on personal and public costs [1]. According to Goldberg and McGee [2], pain is considered as a global public health priority because it affects all populations in different ways (acutely, chronically, or intermittently), without distinction. Moreover, pain is in epidemic proportions and often remains unrelieved [3]. Once pain management becomes a basic human right, being considered as negligence when it is unreasonably untreated [4], it will trigger many initiatives in medical research to continuously develop new analgesics. Chronic pain claims emergent concerns because it faces a public health problem reflected by misuse, abuse, opioid addiction, and overdose deaths [5]. In the past decades, substantial advances in our understanding of pain’s molecular underlying mechanisms have been achieved. There are potential targets identified and investigated focusing on the development of novel analgesic drugs with higher efficacies and less adverse effects. Of note, several ion channels, mainly the voltage-gated calcium channels (VGCCs) [6], represent interesting pharmacological targets for pain management, for which there is compelling evidence indicating the beneficial effects of selective blockers. 

The activation of peripheral nociceptors starts pain signaling, causing electric excitability from the primary afferent nerve fibers to the sensory neurons in the dorsal root ganglia (DRG) and thereafter the spinal cord and brain. According to Lindsay et al., [7] the peripheral nervous system and spinal mechanisms are the main targets of attention on pain research and drug development due to the simplicity of this approach aiming at afferent nociceptors versus the pharmacological challenges related to the interference on brain targets. The pain transmission process relays on the action potential, membrane depolarization, activation of VGCCs, calcium influx, and release of excitatory neurotransmitters (mainly glutamate, substance P, and calcitonin gene-related peptide–CGRP) [8]. Thus, the VGCCs regulate the calcium influx, which is a pivotal step to propagate pain transmission [9]. The upregulation of these channels was studied and detected in several models of pain along with inflammation, whereas its blockade or deletion is related to analgesia. 

The VGCCs are classified in low- (T-type or Cav3) or high-voltage activation (L or Cav1, N, P/Q, and R-type or Cav2). They can be further subclassified by structural similarities (channel-forming α1-subunit) where L-(Cav1.1, Cav1.2, Cav1.3, and Cav1.4), P/Q-(Cav2.1), N-(Cav2.2), and R-(Cav2.3) channels form heteromultimers (along with auxiliary β-, α2δ, and γ-subunits) and T-type (Cav3.1, Cav3.2, and Cav3.3) channels, which are α1-subunit monomers [10]. It is well known that the dysregulation and maladaptive changes in the expression and physiological functions of any of the VGCCs in the sensory pathway reflect pain-inducing conditions. Although, N- and T-type are the most studied channels as pharmacological approaches to treat pain due to their localization, the other channels show scarce data related to their role in nociception [8]. Additionally, these channels modulate pain along with other mechanisms such as opioid [11] and cannabinoid receptors [12,13] as well as the transient receptor potential (TRP) [14], among others. 

Pain perception is a sensory and emotionally unpleasant experience; moreover, it represents a huge personal, medical, and economic burden that pharmacotherapy targeting brain pathways is now being researched for and developed in the medical field. Obviously, acute pain does not carry the load of chronic pain that is conceived as a disease on its own and as secondary to an underlying disease (like a symptom). Chronic pain is related to neuronal adaptations and is high risk for psychological distress and sleep deprivation, among other consequences impairing the quality of life. Given that the VGCCs represent a promise in pain therapeutics, the present review aims to summarize the main advances in the current development of drugs for clinical use and selective ion channel blockers investigated in the pre-clinical field. 

## 2. L-Type Channels 

Additionally, known as the dihydropyridine channels, these L-type (long-lasting) channels exist in four different isoforms, namely Cav1.1–Cav1.4, given that Cav1.2 and Cav1.3 are more expressed in the nervous system in mammals [15]. More specifically, in the dorsal horn of the spinal cord, Cav1.3 are found in dendrites, while Cav1.2 are restricted to the soma, but both are believed to control neuronal hyperexcitability by regulating neuronal firing postsinaptically [16]. Originally, the Cav1.x research focused on the cardiovascular system, and posteriorly the interest shifted to other disease states [17]. 

The dysregulation of Cav1.2 and Cav1.3 in dorsal horn neurons would be responsible, respectively, for short-term (the increase in nociceptive pathway responsiveness) and long-term (plasticity) sensitization linked with neuropathic pain [18]. In neuropathic pain models, Cav1.2 and Cav1.3 isoforms would be downregulated in DRG, whereas Cav1.3 is found to be upregulated in the spinal cord [19]. Besides, knocking down the Ca1.2 decreased behavioral hypersensitivity and reversed the neuronal hyperexcitability in the dorsal horn [20]. In a recent review, Roca-Lapirot et al. [21] summarized that acute sensitization, hyperexcitability induced by inflammation, and allodynia or hyperalgesia induced by neuropathy can effectively be reversed when different families of calcium channels blockers such as nimodipine (dihydropyridines), verapamil (phenylalkylamines), or diltiazem (benzothiazepines) were intrathecally delivered. Li and colleagues also associated the increased expression and activity of Cav1 channels in DRG, especially Cav1.2, with sleep-deprivation-mediated persistent postoperative pain [22]. In this pain model, the authors revealed that intraperitoneal injection of nimodipine showed antinociceptive effects, thus corroborating the findings of Wong and colleagues by using the tail-flick test [23] or by Kawashiri et al. [24] when testing orally treated oxaliplatin-induced neuropathic rats receiving nimodipine. Alternatively, other authors showed that spinally delivered verapamil, diltiazem, and nimodipine did not show analgesic effects in the neuropathic pain models [25,26,27].

The pathomechanisms of migraine are related to the activation of the trigeminovascular system (TGVS), among other circuits, where cortical spreading depression (CSD) outcomes in aura preceded migraine episodes [28]. In this sense, if the Cav1 channels are shown to be upregulated in mouse brains subjected to episodes of CSD, then they might be involved in the pathophysiology of the condition [29]. The in vitro data indicated that the blockade of L-type VGCCs by nimodipine decreased the potassium-induced CGRP release in rat dura mater [30]. Nimodipine also caused a partial relaxation of the rat basilar artery, as well as propranolol, via blocking L-type channels [31], which might reflect the effects of L-type VGCC blockers in migraine. Clinically, studies using flunarizine (a selective calcium channel antagonist) and nimodipine showed a reduction in the frequency of the headache attacks [32,33]. 

Overall, there is even strong evidence that L-type VGCCs are involved in pain transmission, although no analgesic effects are verified in human yields [21] and no consistent data from animals were attributed. It is suggested that the contrasting findings in the animal models could be attributed to the selectivity to the L-type channel subtype [8] or the mode of application of the antagonists/blockers [21]. It is important to remind that dihydropyridines demonstrate their electrophysiological specificity by mainly reducing vascular resistance, phenylalkylamines are less voltage dependent, and benzothiazepines could have their sensitivity modified by alternative splicing in Cav1.2 channels in blood vessels [10,34]. Therefore, the limitation to efficient treatments for pain would be based on the prominent expression of these channels in the cardiovascular system [35], which needs to be circumvented by being directly delivered to the central nervous system [21]. Along with that, a drug mainly targeting spinal Cav1.2 or Cav1.3 is difficult and challenging for drug development because of their different biophysical properties and significant sequence homology [10].

## 3. P/Q-Type Channels 

P/Q-type (from Purkinje/Cerebellar) or Cav2.1 mediate P/Q-type Ca^2+^ currents and are included in the Cav2 family of the VGCCs [36]. These channels show an important role in controlling neurotransmitter release, once they are pre-synaptically located at the axon terminals and somatodendritic compartments of neurons [37,38], mainly in descending facilitatory systems from the rostral ventromedial medulla, which contributes to tactile allodynia in peripheral neuropathy [39]. Additionally, in streptozotocin-induced diabetic neuropathy, there was an increase in the expression of P/Q-type VGCCs in small- and medium-size neurons from DRG (ascending pain pathway) [40].

Luvisetto and colleagues [41], using Cav2.1α1 null mutant mice, found that these animals exhibit pronociceptive responses in inflammatory and neuropathic models but have an antinociceptive response against noxious thermal stimuli. Another example is the result of Fukumoto et al. [42] that highlighted that P/Q-type VGCCs would be pronociceptive once mice with a spontaneous mutation in those channels (producing lower voltage sensitivity of activation) show hypoalgesic responses demonstrated by a lower sensitivity to thermal (tail flick) but also mechanical (von Frey) and chemical (intraplantar formalin) stimuli.

The P/Q-type VGCCs are highly homologous to the N-type VGCCs given that the last one is the preferred target for pain therapeutic once there is no success in optimizing the specific P/Q-type VGCC blockers. Thus, small molecules are mixed N-P/Q-type VGCC blockers in pharmaceutical development [43]. Most of the discovered animal venom toxins are not specific to P/Q-type VGCCs, except by ω-agatoxin IVA and ω-agatoxin IVB derived from *Agelenopsis aperta* spider venom [43,44]. The blockade of the P/Q-type VGCCs by spinally delivered agatoxin IVA and agatoxin TK (its related peptide) display antinociceptive effects in acute inflammatory models in rodents [45,46,47,48,49]. However, there are controversial studies demonstrating that the agatoxin IVA could inhibit [50] or not inhibit pain in neuropathic pain models [27,51]. Another toxin recently studied for its analgesic effects is Tx3-3, derived from the venom of the *Phoneutria nigriventer* spider. The purified fraction blocks were preferentially P/Q-type and R-type VGCCs [52], and, in vivo, they were demonstrated to be efficient in murine models of noxious thermal stimuli (tail flick), neuropathic (partial sciatic nerve ligation and streptozotocin-induced diabetic neuropathy), and inflammatory pain [53,54], along with the fibromyalgia model [55]. 

So far, Cav2.1 are critically suggested to be important in genetic studies of familial hemiplegic migraine [56,57,58]. In this instance, although the increased calcium influx through P/Q-type VGCCs contributes to the cortical excitability being triggered to spread depression (aura putative mechanism), a decreased calcium current through Cav2.1 in the periaqueductal grey via a channel blockade facilitates trigeminal nociception [59,60,61]. Through this perspective, Inagaki et al. [62] studied a Cav2.1 modulator named tert-butyl dihydroquinone (BHQ), which, in a heterologous system transfected with a familial hemiplegic migraine mutation, was demonstrated to slow deactivation and inhibit the voltage-dependent activation of P/Q-type VGCCs. All in all, the BHQ effects corroborate the understanding of Cav2.1’s role in the migraine mechanism. 

## 4. N-Type Channels

N-type (from Neuronal, non-L) or Cav2.2 were found in the dendritic shafts and presynaptic terminals of central and peripheral neurons [63,64]. They are the main contributors to the nociceptive signal transmission in the dorsal horn of the spinal cord, once there is an up-regulation in nociceptive neurons [65]. The inhibition or deletion of these channels evidenced their role during pain states (see review by Hoppanova and Lacinova [66]). Selective and specific Cav2.2 blockers are being studied in vitro or were tested in rodent pain models, and they include several ω-conotoxins derived from marine cone snails as GVIIA from *Conus geographus;* MVIIA and MVIIB from *Conus magus*; SVIA and SO-3 from *Conus striatus*; CVIE from *Conus catus*; FVIA from *Conus fulmen*; RVIA from *Conus radiatus*; TVIA from *Conus tulipa;* MoVIA and MoVIB from *Conus moncuri*; RsXXIVA from *Conus regularis*; and Eu1.6 from *Conus eburneus* (see the recent reviews by Ramirez et al. [67] and Trevisan and Oliveira [68]). Similarly, the toxins purified from the venom of the Chinese bird spider *Ornithoctonus huwena* showed the N-type blocker properties in electrophysiologic studies such as HWTX-X [69] and HWTX-XVI, but only the last toxin was tested in vivo for eliciting analgesic responses in the formalin-elicited pain model [70].

The toxins with Cav2.2 blocking action might also act in other channels to produce analgesia. Examples that are being studied include: GeXIVA from *Conus generalis* that also act in inwardly rectifying K^+^ currents (GIRK) [71,72]; Vc1.1 from *Conus victoriae*; and RgIA from *Conus regius* to target the α9-subunit-containing nicotinic acetylcholine receptors (α9-nAChR) as well as the GABA_B_ receptor mechanisms [73,74,75] (both in Phase II clinical trials); and Cd1a from the venom of the spider *Ceratogyrus darlingi* that interferes with Cav2.2 inactivation and the α-subunit pore, while altering the activation gating of Nav1.7 [76]. 

Although many peptides are currently being tested in preclinical models, only MVIIA (SNX-111) was synthetically produced, and it is clinically approved and marketed under the name Ziconotide (Prialt^®^) to be used in refractory pain via intrathecal administration, even though the main concerns are its challenging side effects [77,78]. GVIA (SNX-124) is in a Phase Ia clinical trial, and it was well tolerated intrathecally [79]. A huge limitation for those drugs is the ability to cross the blood–brain barrier, and ziconotide dosed systemically can produce severe hypotension in rodents, by inhibiting sympathetic neurotransmission [80,81]. Thus, CVID and its synthetic form AM336 (leconotide) were reported with promising perspectives even in animal experiments via the intravenous route [82]. Notwithstanding, Cav2.2 activity seems to be related to heat but not mechanical hypersensitivity in peripheral axon terminals innervating skin, where intraplantar ziconotide reduced the sensitivity in the capsaicin-induced pain [83]. However, recently, Hasan et al. [84] showed that MVIIA injected into the mouse paw could alleviate the mechanical allodynia, lacking effectiveness against the thermal paw withdrawal threshold on mice with postsurgical and chemotherapy-induced neuropathic pain (oxaliplatin intraplantar), but was ineffective on the cisplatin model.

HVA (high-voltage-activated) VGCC activity depends on the interaction between alpha and beta subunits [85]. Then, small molecules such as IPPQ (a quinazoline analog), which selectively target that interface on Cav2.2, show potential for pain therapeutics. That molecule inhibited the N-type VGCC currents in sensory neurons and their pre-synaptic localization and spinal neurotransmission in vivo, thus resulting in decreased neurotransmitter release and reduced mechanical allodynia and thermal hyperalgesia in murine models of postsurgical and neuropathic pain [86,87]. That promising molecule displayed superior effects in comparison with others with the same target, such as BTT-369 (a benzoylpyrazoline analog) [88], based on its different kinetics and toxicity profile [86].

Searching for analogues of conotoxins, fluorophenoxyanilide derivatives were screened, tested, and revealed an improved activity to block N-type VGCCs [89]. However, no more studies were found regarding these compounds.

## 5. Mixed-Target Toxins 

Most animal toxins have more than one VGCC target, and they have been proved to show analgesic effects in vivo (Table 1). The different therapeutical potentials depend on the affinity and selectivity for a subtype channel and how the peptide binds to the channel. Once more, for the Conus-derived toxins studied so far, the majority of conotoxin patents have more academic than industrial value [90]. However, the *Phoneutria nigriventer* venom spider displays a variety of peptides with analgesic effects in rodent pain models [91]. To be highlighted is the PhTx3-6 from the venom fraction PhTx3. The peptide also known as Phα1β was produced recombinantly in *Escherichia coli* and called CTK 01512-2, exhibiting the same efficacy as the native form in different pain animal models (see the reviews by Antunes et al. [92] and da Silva et al. [93]). For instance, the native toxin Phα1β displayed marked analgesic effects in the mouse model of cyclophosphamide-induced hemorrhagic cystitis when dosed intrathecally [94]. Moreover, the recombinant toxin CTK 01512-2 exhibited analgesic actions in mice with experimental autoimmune encephalitis, when administered by either intrathecal or intravenous routes [95]. Its main target would be N-type VGCCs, but it could inhibit R- and P/Q-type VGCCs with a comparable affinity [96]. Less investigated, though, but no less interesting, are the fractions PhTx3-4 and PhTx3-5 currently being investigated, with potential favorable preclinical effects.

## 6. Mixed-Target Small Molecules

Lomerizine is a dual antagonist of L- and T-type VGCCs that was studied in migraine clinical trials [101] but was withdrawn in Phase II [102]. Its derivatives resulted in N-type VGCC blockers with higher affinities and selectivity, displaying analgesic activity in pre-clinical models [103,104,105,106,107]. However, further studies cannot be found. 

The orally available 2-aryl indoles were identified as potent N-type VGCC blockers, but did not have good cardiovascular profiles. Thus, a modification in the molecule substituting N-triazole oxindole (origin of TROX-1) led to analgesic properties in inflammatory and neuropathic pain, without cardiovascular impairment [108,109,110], but it had no selectivity for the blockage of P/Q-, R-, or N-type VGCCs [111]. Another small oral non-peptide that is selective to N-type VGCCs is ZC88, which is a 4-aminopiperidine derivative [112,113]. The molecule potentiated morphine analgesia and attenuated morphine tolerance and dependence in rodents [112]; additionally, it displayed analgesic activity in inflammatory pain, with a synergistic action when combined with ibuprofen [114]. 

Dihydropyridine derivatives such as amlodipine and cilnidipine exerted marked analgesic effects by blocking mixed N- and L-type VGCCs when administered intrathecally or orally in animals [115,116]. Cilnidipine was evaluated intrathecally in a neuropathic pain model and maintained its antinociceptive properties [117,118]. Pranidipine was efficient at reducing the licking time in the formalin test, also by inhibiting N-type VGCCs [119].

Phytochemicals such as the physalin F were demonstrated to act in N- and R-type VGCC, decreasing the tactile hypersensitivity in neuropathic pain models [120]. Additionally, isolated iridoids glycoside compounds reduced nociception by acting through Cav2.2 or Cav3.2 VGCCs [121,122].

## 7. T-Type Channels

The T-type (from transient) or Cav3 VGCCs family behaves as a single α1 subunit, with no need for auxiliary subunits [123]. They are classified as LVA being activated by subthreshold stimuli and are represented by three different α1 subunits named Cav3.1, Cav3.2, and Cav3.3 VGCCs [124]. These channels can control neuronal excitatory transmission in primary afferent neurons, contributing to nociceptive circuitry [125,126]. Regarding pain, the specific contribution of Cav3.1 and Cav3.3 need to be further evaluated, even if there is strong evidence that Cav3.2 is upregulated in pain states with differences in rodents, humans, and non-human primates (see recent review Harding and Zamponi [126]). 

To modulate pain perception using T-type VGCCs, many blockers that are approved for the therapy of other diseases were studied, but others were developed for academic research focusing on a pan-blockade or specific subtype of Cav3. NNC 55-0396 is a structural analog of mibefradil (effective antihypertensive) that inhibits both T-type and HVA VGCCs with higher selectivity for Cav3.x [127]. Preclinically, both NNC 55-0396 and mibefradil were shown to prevent and reduce visceral nociceptive behavior [128], cystitis-related bladder pain [129], and paclitaxel-evoked neuropathic pain in rodents [130]. However, there are mixed results indicating that the analgesic effect of mibefradil depends on the pain model and the route of administration, according to Pexton et al. [102], which is justified by the fact that mibefradil does not cross the blood–brain barrier [131]. 

Dihydropiperidines are considered L-type VGCC blockers, but their derivatives have been described as Cav3.x inhibitors, being able to reduce pain in mice [132,133], as well as dihydropyrimidine derivatives [134]. Clinically existing medicines such as bepridil (a diamine used as anti-arrhythmic) and pimozine (diphenylbutylpiperidine used as antipsychotic) were also shown to block T-type VGCCs and decrease nociception in animals with colonic and bladder pain [135]. Still regarding piperidine derivatives, TTA-P2 is a recently synthesized selective and potent Cav3 inhibitor that was demonstrated to reduce pain responses in mice in acute inflammatory pain and diabetic neuropathy [136,137,138]. In the same perspective, TTA-A2 efficiently inhibits Cav3, demonstrating higher potency for Cav3.2, and it was shown to decrease pain in an irritable bowel syndrome model [139], nocifensive visceromotor responses to noxious bladder distension [140], and bortezomib-induced peripheral neuropathy [141]. Z944 is a piperazine-based and T-type selective antagonist that was also proven to decrease nociception in inflammatory acute and chronic pain states [142] as well as in a trigeminal neuralgia model [143] when injected systemically. Clinically, Z944 proved to be safe in healthy males and reduced the pain sensation score from both capsaicin and UV-irritated skin models [144]

Being slightly more investigated, ethosuximide (a dicarboximide used as an antiepileptic drug) was revealed to be antinociceptive in chemotherapy-induced pain [145]; however, it failed in a clinical trial investigating its analgesic effects in patients with neuropathic pain [146]. Despite this, ethosuximide is still being tested in clinical trials to verify its possible effects in the abdominal pain related to irritable bowel syndrome [147,148] and non-diabetic peripheral neuropathic pain [149]. Ethosuximide showed limited efficacy when compared to zonisamide (sulfonamide anticonvulsant agent with Cav3.x blocker properties [150,151]), in neuropathic mechanical hypersensitivity [152]. Zonisamide has been shown to reduce pain behavior in several pain models [153,154] and this compound was also effective to treat severe and intractable central poststroke pain in two patients [155], migraine [156,157,158,159], and painful diabetic neuropathy [160]. ABT 639 is also a sulfonamide and nonselective Cav3 antagonist that can produce antinociception when administered intraperitoneally in murine models of neuropathic but not inflammatory pain [161,162], or intrathecally in an inflammatory bowel disease model [163]. Nevertheless, it failed in treating pain in human clinical trials in patients with diabetic peripheral neuropathy [164,165]. 

Regarding the peptides or neurotoxins with T-type potent blocker properties, kurtoxin isolated from the venom of the scorpion *Parabuthus transvaalicus* [166] and ProTx-I and ProTx-II from the venom of tarantula *Thrixopelma pruriens* [167] need to be mentioned, albeit they also modulate HVA VGCCs and Nav. The recently documented Tap1a and Tap2a are tarantula venom peptides isolated from *Theraphosa apophysis* that can modulate the activity of both NaV and CaV3 channels. Both peptides were demonstrated to reduce the colonic mechanical pain in a mouse model of irritable bowel syndrome [168]. Alpha-cobratoxin (α-CTx) from the *Naja naja siamensis* (Thailand cobra) and receptin (its chemically modified toxin) have been revealing potential antinociceptive effects via intraperitoneal or intracerebroventricular administration in mice, as assessed in different pre-clinical pain states [169,170,171,172,173,174]. Its mechanism is believed to be through the activation of the muscarinic M4 receptor, inhibiting T-currents via the G-protein [175]. 

Cannabinoids such as anadamide [176], N-arachidonoyl dopamine (NADA) [177], some lipoamino acids [13], synthetic cannabinoid receptor agonists [178], cannabidivarin and cannabigerolic acid [179], and delta-9-tetrahydrocannabinol (THC) and cannabidiol (CBD) [180] show marked inhibition of all T-type channels. Antinociceptive properties were seen when lipoamino acids [13], the terpenes camphene and alpha-bisabolol [181] derived from cannabis plants, CBD [182], and mixed T-type/cannabinoid receptor ligands [183,184,185] were tested in vivo and revealed as dependent on Cav3 VGCCs, mainly Cav3.2, to produce analgesia. 

## 8. R-Type Channels

Distributed in the peripheral and CNS, mainly over the cell soma and proximal neuronal dendrites [65], Cav2.3 or R-type (from residual) VGCCs are encoded by the CACNA1E gene (VGCC subunit alpha1 E, see the recent review by Scheinder et al. [186]). They are engaged in visceral nociception control; once Saegusa et al. [187] revealed that mice lacking the α1E subunit had altered pain responses, and Yang and Stephens [188] showed that partial-sciatic-nerve-ligation-induced neuropathy in the DRGs of these mice triggered drug resistance through alternative adaptive mechanisms. Alternatively, the endogenous amino acid l-cysteine modulated R-type VGCCs and increased the nociceptive behavior in an acetic acid visceral pain model in wild type mice when administered subcutaneously, intraperitoneally, or intrathecally, but had no effect in Cav2.3 knockout mice [189], proving Cav2.3’s role in visceral pain. 

As a selective Cav2.3 inhibitor, SNX-482 from the venom of the *Hysterocrates gigas* spider was demonstrated to decrease R-type currents when administered intrathecally and to diminish the nociception in a model of neuropathic pain [190]. Another current marine toxin investigated is contulakin g (CGX) from *Conus geographus*, which has a mechanism of action that is dependent on the R-type VGCCs in sensory neurons and is unveiled by intrathecal injection in rodent models of inflammatory and neuropathic pain [191].

## 9. α2δ Subunit Blockade

An increased expression of the Cavα2δ auxiliary subunit of VGCCs was observed in rodents with neuropathic pain [192,193], which could contribute to enhanced pre-synaptic excitatory neurotransmitter release and related behavioral pain [194,195]. In this scenario, spinally, Cavα2δ differentially modulates N-, L-, and P/Q-types to promote behavioral hypersensitivity [193], but the selective and high-affinity binding in Cavα2δ inhibits abnormal neuronal activity [196] in a sufficient manner to provide analgesic action.

Gabapentinoids such as pregabalin, gabapentin, and mirogabalin (used to prevent and control seizures) have demonstrated therapeutic efficacy to treat pain clinically and pre-clinically (see the recent review by Chen et al. [197]). Although their effects can be attributed to a multitude of mechanisms, they downregulate the VGCCs and N-methyl-D-aspartate (NMDA) receptor expression and decrease excitatory neurotransmitter release by inhibiting the Cavα2δ of HVA VGCCs [198,199]. Mirogabalin is the current novel gabapentinoid being explored in the pain field. The drug shows promising results to treat pain conditions with good efficacy and tolerability [200]. 

## 10. Conclusions

Therapies focusing on new pain treatments are in development and are constantly being improved using diverse research tools. Computational modeling aids in screening for promising molecules, but the efficacy and safety need to be established in animal models, which could still differ between rodents and humans. The potential effects of VGCC blockers were demonstrated in a variety of pain models reflecting the clinical scenario, and VGCCs have undoubtedly proved their potential for pain management (Figure 1). This narrative review summarizes the controversial outcomes regarding L-type VGCCs in analgesia. In addition, P/Q VGCCs inhibitors are mainly related to migraine studies, and R-type VGCCs are the least studied due to the lack of selective blockers. It can be highlighted that N- and T-type VGCC blockers are extensively documented and investigated, condensing the majority of pain research. Whereas several toxins derived from animal venom (marine conus, spiders, and scorpions) are VGCCs inhibitors, the majority are limited to in vitro studies, and less than half were tested as antinociceptive in rodent pain models, with only ziconotide being currently marketed to treat severe pain. From another perspective, gabapentinoids such as gabapentin and pregabalin pursue high Cavα2δ blocker properties and are also marketed and approved for pain treatment, even though they were not primarily designed as analgesics. Lately, in the current spotlight, mirogabalin has stood out for its potential in clinical trials; however, it has been not yet been approved by the US Food and Drug Administration. In general, the mixed-target compounds in development have been explored and revealed encouraging results. It is seen that non-opioid strategies are overlooked for relieving pain. Ongoing research focusing on natural products offer promising benefits and are valuable strategies due to their marine biodiversity, not only regarding animal venom but also seaweeds with compounds showing analgesic properties, which have not yet been exploited.

## Figures and Tables

**Figure 1 ijms-24-09223-f001:**
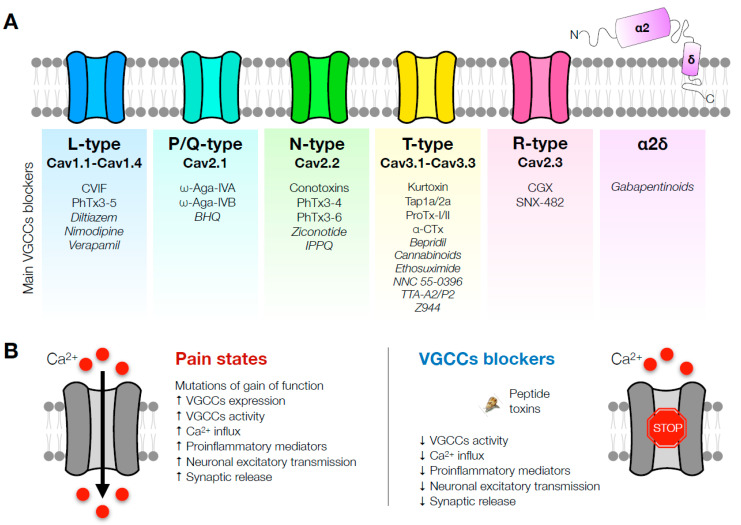
Graphical conclusion. Mechanisms of involvement of VGCC blockers in pain states. (**A**) Main VGCCs blockers according to their VGCCs type. (**B**) Summary of the underlying mechanisms of current naturally based and synthetic VGCC blockers on pain control.

**Table 1 ijms-24-09223-t001:** Main mixed-VGCC target peptides derived from venom with analgesic properties.

Peptide	Channels Target	Organism Derived	Analgesic Effect in Pain Rodent Models—Reference
GVIA	Cav2.1, and Cav2.2	*C. geographus*	Fukuizumi et al. [25] Murakami et al. [49] Scott et al. [97]
MVIIC	Cav2.1, and Cav2.2	*C. magus*	Dalmolin et al. [53]
SVIB	Cav2.1, and Cav2.2	*C. striatus*	-
CVIA	Cav2.1, and Cav2.2	*Conus catus*	-
CVIB	Cav2.1, and Cav2.2		-
CVIC	Cav2.1, and Cav2.2		-
CVID	Cav2.1, and Cav2.2		Scott et al. [97]
CVIF	Cav1., Cav1.3, Cav2.2, and Cav2.3		Berecki et al. [98]
CnVIIA	Cav2.1, and Cav2.2	*Conus consors*	-
PhTx3-4	Cav2.1, and Cav2.2	*P. nigriventer* spider	Da Silva et al. [99]
PhTx3-5	Cav1.x, and Cav.2.2		Oliveira et al. [100]
PhTx3-6	Cav2.1, Cav2.2, and Cav2.3		See review by da Silva et al. [93]

## Data Availability

The authors confirm that the data supporting the findings of this study are available within the article.

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
