# Peer review of "Current Drug Development Overview: Targeting Voltage-Gated Calcium Channels for the Treatment of Pain"

_ijms, 2023, doi:10.3390/ijms24119223_

Round 1

Reviewer 1 Report

This paper brings an overview of naturally based and synthetic VGCC blockers, it is a well organized paper, please improve it according to the following suggestions.

 1. For the convenience of readers, please add one figure to make a summary on the underlying mechanisms of current naturally based and synthetic VGCC blockers on pain control with more details.

2. The current manuscript needs to be polished. 

Minor  improvement

Author Response

We thank the reviewer for their commitment to improving the manuscript's quality. 

1) We have improved the figure in the conclusion section according to the suggestion. However, the underlying mechanisms between synthetic and natural ligands are based on the same resultant effects which we added in the figure. 

2) Some language typos were corrected to improve readability as suggested by reviewer #2 also.

Reviewer 2 Report

Dear Ladies and Gentlemen, Dear Journal-Team,

the manuscript 'Current drug development overview: targeting voltage-gated calcium channels for the treatment of pain' summarizes the knowledge of the different calcium channels involved in pain transmission. It is well written. The table and the figure are sufficient.

a) Please describe all mentioned pharmaceutical abbreviations, and describe the calcium channels in more detail (L-type=long lasting, N-type=neuronal, non-L, P-type=Purkinje, Q-type=cerebellar, R-type=residual, T-type=transient, Cav1=slowly inactivating, HVA=high volatage activated in section N-type channels, second paragraph from below, CACNA1E=calcium voltage-gated channel subunit alpha1 E in section R-type channels).

b) Language: 1. Section L-Type channels, line 3, please change to: 'in the dorsal horn of the spinal cord...are restricted to the soma'.

2. Section L-type channels, line 12: 'cord. Besides'.

3. Section L-type channels, line 13: 'in the dorsal horn'.

4. Section N-type channels, line 3: 'in the dorsal horn of the spinal cord'.

5. Section N-type channels, line 30: 'via the intravenous route'.

6. Section N-type channels, line 36: correct the punctuation after 'interplantar'.

7. Section Mixed-target toxins, line 2: 'it was withdrawn'.

8.Section T-type channels, line 47: 'but'.

9. Table 1: 'Main mixed-VGCCs target peptides derived from venom with analgesic properties.'

c) Please check the references according tp the Journal Style Guidelines: In Reference 3 (Sessle) the year of publication is mentioned twice, and check the uniform usage of small or capital letter in the article title throughout the reference list.

Sincerely, 

please refer to the above mentioned comments

Author Response

We thank the reviewer for their commitment to improving the manuscript's quality. 

a) The mentioned abbreviations were included as well as the details required.
b 1-9) We have checked the language mistypos accordingly in the notes. 
c) The references were generated through Endnote following the Journal Instructions for Authors, but we have amended the capitalization of the words.